# Genomic selection for salinity tolerance in *japonica* rice

Jérôme Bartholomé[1,2,3]*, Julien Frouin[2,4], Laurent Brottier[2,4], Tuong-Vi Cao[2,4], Arnaud Boisnard[5], Nourollah Ahmadi[2,4], Brigitte Courtois[2,4]

**1** UMR AGAP Institut, CIRAD, Cali, Colombia, **2** UMR AGAP Institut, Institut Agro, Univ Montpellier, CIRAD, INRAE, Montpellier, France, **3** Alliance Bioversity-CIAT, Recta Palmira Cali, Colombia, **4** CIRAD, UMR AGAP Institut, Montpellier, France, **5** Centre Français du Riz, Arles, France

* jerome.bartholome@cirad.fr

**Data Availability Statement:** The link for the data availability statement is the following: https://doi.org/10.18167/DVN1/O1AYGP.

## Abstract

Improving plant performance in salinity-prone conditions is a significant challenge in breeding programs. Genomic selection is currently integrated into many plant breeding programs as a tool for increasing selection intensity and precision for complex traits and for reducing breeding cycle length. A rice reference panel (RP) of 241 *Oryza sativa* L. *japonica* accessions genotyped with 20,255 SNPs grown in control and mild salinity stress conditions was evaluated at the vegetative stage for eight morphological traits and ion mass fractions (Na and K). Weak to strong genotype-by-condition interactions were found for the traits considered. Cross-validation showed that the predictive ability of genomic prediction methods ranged from 0.25 to 0.64 for multi-environment models with morphological traits and from 0.05 to 0.40 for indices of stress response and ion mass fractions. The performances of a breeding population (BP) comprising 393 *japonica* accessions were predicted with models trained on the RP. For validation of the predictive performances of the models, a subset of 41 accessions was selected from the BP and phenotyped under the same experimental conditions as the RP. The predictive abilities estimated on this subset ranged from 0.00 to 0.66 for the multi-environment models, depending on the traits, and were strongly correlated with the predictive abilities on cross-validation in the RP in salt condition (r = 0.69). We show here that genomic selection is efficient for predicting the salt stress tolerance of breeding lines. Genomic selection could improve the efficiency of rice breeding strategies for salinity-prone environments.

## Introduction

Soil salinization is a major challenge worldwide, affecting about 20% of all irrigated land [1]. The predicted rise in sea level due to climate change will increase the percentage of salt-affected land, especially in coastal and delta areas [2]. Salinity is one of the most important factors reducing rice (*Oryza sativa* L.) crop productivity in river deltas, including some of the major production areas in Asia and Europe. This is a major concern for rice growers, as rice is considered to be more salt-sensitive than other cereals, such as wheat or barley [3]. Indeed, salt stress has a strong effect, decreasing growth or survival at the seedling stage in rice, even at moderate salinity levels, such as 3.4 dS·m$^{-1}$ [4]. Salt stress also significantly impacts rice grain

**Funding:** This research was conducted under the framework of the FACCE-JPI project GreenRice (Sustainable and environmental-friendly rice cultivation systems in Europe) and was funded by the French National Agency for Research (ANR-14-JFAC-0005-01). The funders had no role in study design, data collection and analysis, decision to publish, or preparation of the manuscript.

**Competing interests:** The authors declare that they have no competing interests.

**Abbreviations:** RP, reference panel; BP, breeding population; SNP, single nucleotide polymorphism; BP41, a subset of 41 lines of the BP; CRTL, control condition; GBLUP, Genomic best linear unbiased prediction; GBS, genotyping by sequencing; G×E, genotype by environment interactions; LA, leaf area; LL, leaf length; PA, predictive ability; QTL, quantitative trait locus; R_S, ratio of root-to-shoot dry weights; RKHS, reproducing kernel Hilbert space; RL, root length; ROOT, root dry weight; RP, reference panel; SALT, salt condition; SHOOT, shoot dry weight; SLA, specific leaf area; SNP, single nucleotide polymorphism; TIL, number of tillers.

yield, even at low salinity levels, with decreases in yield components, including tiller number, spikelet number, and grain weight [5]. Sensitivity depends not only on the intensity of the stress, but also on its timing relative to plant development. Like other crops, rice is more sensitive during the seedling and reproductive stages [6]. Efforts have been made to unravel the mechanisms involved in salt tolerance in rice at the physiological, molecular, and genetic levels [7,8]. Three mechanisms have been implicated in plant salt-stress tolerance: ion exclusion ($Na^+$ or $Cl^-$), tissue tolerance, and osmotic tolerance [6,9]. Phenotyping is challenging due to the complex interactions between environmental and genetic factors, and accurate protocols are required to quantify plant responses to salt stress [6].

Since the start of the Green Revolution, rice breeders and geneticists have been screening the diversity of *O. sativa*, with the aim of identifying tolerant genotypes [10,11]. Screening protocols have been developed to evaluate lines subjected to different stress levels at the seedling stage [12]. In the 1970s, for example, the International Rice Research Institute in the Philippines evaluated about 100,000 accessions for salt tolerance by visual damage scoring. Less than 20% of the lines tested were found to be tolerant, suggesting that salt tolerance is not a major characteristic in rice [13,14]. Most of the tolerant accessions belonged to the *O. sativa* subspecies *indica* (e.g., Nona Bokra, Pokkali, Hasawi, Getu, Cheriviruppu, Ddamodar, Solla and Ketumbar) and many originated from mangrove areas, but some tolerant lines were also found in the *japonica* subspecies (e.g., Honduras, Slava, and Gigante Vercelli, [15,16]). These tolerant landraces or ecotypes were subsequently used as donors for salinity tolerance genes in breeding programs [13]. With the advent of molecular markers, tolerant landraces were used in various types of crosses, to map quantitative trait loci (QTLs) and to shed light on the genetic control of salinity tolerance [17–19]. Several hundred QTLs and dozens of genes have been identified, highlighting the complex genetic control of this trait [12,13]. These QTLs and genes are spread throughout the rice genome, but with a higher concentration on chromosome 1. The major QTL *SalTol*, located on chromosome 1, was used for marker-assisted selection [20–23]. Most efforts have focused on pre-breeding activities based on QTL introgression, and a number of salt-tolerant varieties, such as IR64-Saltol, ASS996-Saltol and BRRI dhan 47, have emerged from these approaches [24,25]. However, most of these varieties belong to the *indica* subspecies, which grows in tropical and subtropical regions. In the framework of the European Neurice project, *Saltol* was introgressed in Spanish, Italian and French temperate *japonica* varieties through marker-assisted selection(http://www.neurice.eu).

Despite major advances in the marker-assisted selection of a major QTL/gene, breeding for salt tolerance is rendered more difficult by the complex genetic architecture of the trait, as hundreds of genes associated with different mechanisms are involved [13,26]. Favorable alleles at several QTLs/genes are therefore required to occur together to confer a significant level of tolerance in field conditions. There is a need to combine QTL/genes controlling salt tolerance at both the vegetative and reproductive stages. In addition, interactions with environmental conditions, such as the timing and intensity of salt stress, play a major role, making it harder to identify the best-performing lines. Indeed, tolerant lines must also perform well under more favorable conditions, and must be adapted to the various environments encountered in farmers' fields [27]. In this context, genomic selection is a potential tool for accelerating genetic improvement for salt tolerance [28]. Genomic selection requires the initial calibration of a prediction model on a training population that has been both genotyped and phenotyped. This prediction model is then applied to candidates for selection, based purely on their genotypes [29]. This approach enables breeders to optimize their breeding strategy by selecting early in the breeding cycle for complex and expensive-to-evaluate traits, thereby decreasing the length of the breeding cycle [30]. In rice, genomic prediction has been successfully evaluated for different traits and different types of populations [31–33]. Most of these studies focused on predicting

performances in normal conditions, but a few have assessed the accuracy of prediction for performance under abiotic stress conditions, such as water deficit in particular [34,35]. Genomic prediction models integrating genotype-by-environment interactions have recently been developed [36–39]. These models tend to increase prediction accuracy, by incorporating multi-environment data and modeling marker-by-environment interactions. They are particularly useful for breeding programs targeting stress-prone environments, as they make it possible to combine performances in the presence and absence of stress, to increase accuracy [31].

In this study, we used material from European rice breeding programs to evaluate the potential of genomic prediction for improving salinity tolerance in *japonica* rice adapted to temperate regions. Due to environmental constraints (low temperatures and long days) almost all of the rice varieties grown in Europe belong to the *japonica* subspecies [40]. A panel of European accessions was recently evaluated for tolerance to salt stress at the seedling stage, revealing considerable variability for various traits (growth parameters and sodium ($Na^+$) and potassium ($K^+$) mass fractions) despite the absence of the favorable allele at the *Saltol* locus [41]. These findings indicate that the use of minor genes and selection for this quantitative variation by genomic selection could help to increase the salt tolerance of temperate *japonica* breeding lines. However, for most breeding programs in temperate regions, only one growing season per year is possible, and the evaluation of salinity tolerance in field conditions is always difficult because of the high level of interannual variability. Genomic prediction models are, therefore, useful for predicting the salinity tolerance of untested genotypes, to make it easier for breeders to take selection decisions as early as possible. The objectives of this study were: i) to assess the accuracy of single and multi-environment genomic prediction models for predicting the performance of accessions from European rice breeding programs for traits related to salt tolerance via cross-validation and ii) to validate these models on an independent subset of breeding lines.

## Materials and methods

### Plant material

We used two different populations (S1 Table). The first population was a reference panel (RP) composed of 241 *japonica* accessions. The RP was previously characterized by Frouin, et al. [41]. These accessions were mostly varieties from temperate regions, with European accessions largely represented (Italy (46.7%), France (12.5%), Spain (12.5%), Portugal (7.1%)), alongside varieties from the United States (10.4%), and more than 15 other countries. The second population was a breeding population (BP) of 393 breeding lines, with 73.8% of the lines derived from the joint breeding program of the *Centre Français du Riz* (CFR, Arles, France) and the *Centre de cooperation international en recherche agronomique pour le développement* (CIRAD, Montpellier, France). This French material included 289 current advanced breeding lines derived from 98 crosses involving 114 parents, 33 of which were included in the RP. The number of lines per cross ranged from 1 to 20, with seven crosses overrepresented (121 individuals). The other lines in the BP were lines from the working collections of European breeders.

### Genotypic characterization of the populations

The two populations were genotyped with the same genotyping-by-sequencing method [42]. The DNA was extracted with the a modified method using hexadecyltrimethylammonium bromide [43], as described by Frouin et al. [41]. The genome was digested with the restriction enzyme ApeKI for library preparation. The libraries were sequenced with a Genome Analyzer II (Illumina, San Diego, California, USA). The Nipponbare reference genome Os-Nipponbare-Reference-IRGSP-1.0, [44] was used for sequence alignment. Single-nucleotide polymorphism (SNP) calling was performed with the Tassel GBS pipeline with the default parameters [45].

Markers with a call rate below 75%, a heterozygosity rate above 10% and a minor allele frequency below 5.0% were discarded. The remaining heterozygotes were converted to missing data. The missing data were then imputed with Beagle v4.0, using the default parameters [46]. The imputed file was split into two sets: the 241 accessions of the RP and the 393 accessions of the BP. Markers with a minor allele frequency below 5.0% were discarded in both populations. This procedure resulted in the identification of 20,255 informative SNPs common to the two populations. Markers in complete linkage disequilibrium were further filtered out: for clusters of markers, only one marker, that with the lowest rate of missing data before imputation and the highest minor allele frequency, was selected to represent the cluster. This filter resulted in 16,993 non-redundant markers, which were used for further analysis. The data can be downloaded from CIRAD Dataverse: https://doi.org/10.18167/DVN1/O1AYGP.

The genetic structure of the 241 accessions of the RP and 393 lines of the BP was assessed by a discriminant analysis of principal components implemented in the R package *adegenet* [47,48]. The functions *find.clusters* and *dapc* were successively used to assign individuals to genetic groups. The number of clusters was set to three, corresponding to the tropical and temperate *japonica* subgroups and admixed accessions. The percentage of the variance used to select the number of axes in the principal component analysis was set at 90%. We also used DarWin v6 software [49] to calculate a simple matching index to assess the dissimilarity between individuals. We also used DarWin v6 software [49] to calculate a simple matching index to assess the dissimilarity between individuals. This index was then used to construct an unweighted neighbor-joining tree. The assignments to groups derived from *adegenet* were projected onto the neighbor-joining tree.

## Assessment of phenotypic performance under controlled conditions

**Reference panel (RP).**　The accessions of the RP were phenotyped for salinity tolerance under hydroponic conditions, as previously described by Frouin et al. [41]. Briefly, the experimental design was a split-plot with three replicates staggered in time. In each replicate, the plants were distributed in 12 tanks (6 control and 6 salt tanks). Two resistant controls (Nona Bokra and Pokkali) and three susceptible controls (IR29, Aychade and Giano) were replicated in each tank. Stress was applied for two weeks, beginning two weeks after sowing. The salt concentration was 50 mM NaCl (3 g/l), corresponding to an electrical conductivity of 6.5 dS·m$^{-1}$. After 28 days sowing, we measured the following growth-related traits in control and salt conditions: the number of tillers (TIL), the lengths of the longest leaf (LL) and the longest root (RL), the length (LGTH) and width (WDTH) of the last fully developed leaf of the main tiller, the dry matter weight for shoots (SHOOT), roots (ROOT) and the last fully developed leaf of the main tiller (LEAF). The Na$^+$ and K$^+$ mass fractions of shoot tissues, expressed as percent of dry matter, were measured on plants grown in salt conditions, by atomic emission spectroscopy (ICP-AES), at the UR59 laboratory at CIRAD Montpellier (ISO9001). We also calculated the following variables: leaf area (LA), calculated as LGTH x WIDTH x 0.75, specific leaf area (SLA), calculated as LA divided by LEAF, the root-to-shoot ratio (R/S), calculated as ROOT divided by SHOOT, and the Na/K ratio, calculated as Na$^+$ mass fraction over K$^+$ mass fraction.

Analysis of variance was performed on the morphological trait data, with a mixed model including salinity/control conditions and genotype as fixed effects and replicate and tank as random effects. The significance of genotype, conditions and genotype x conditions interaction effects were assessed. The least square mean values of the genotypes were calculated with SAS software (Cary, NC, USA).

From the least square mean values for morphological traits, stress response indices were computed as *iTRAIT = (salt−control)×100/control*.

**Subset of the breeding population.** The same methodology was used to phenotype a subset of 41 lines selected from the BP (BP41) in a separate experiment. The design was a split plot with two conditions (control and salt stress) and three replicates. The resistant and susceptible controls varieties were as described above. The same statistical model as for the RP was used for the variance analysis except that, in this case, because of the smaller size of the experiment, the tank and replicate effects were combined. The process used to select the 41 lines is explained below in the "Evaluation of predictive ability" section.

## Statistical models for genomic prediction

The genomic BLUP (GBLUP) and the reproducing kernel Hilbert space (RKHS) models were used to predict breeding values with molecular markers. GBLUP is one of the most popular and robust methods for genomic prediction [50,51]. For GBLUP, we calculated the kernel matrix as follows $K = XX'/p$, $X$ being the centered genotype matrix. $X$ is of dimension $n{\times}p$, where $n$ is the number of genotypes and $p$ the number of markers. For the RKHS model, we used a Gaussian kernel $K(x_i, x_j) = \exp(-h\| x_i - x_j \|^2)$ to calculate the kernel matrix between the marker genotype vectors $x_i$ and $x_j$, where $(i,j){\in}\{1,...,N\}^2$. We estimated the bandwidth parameter $h$ as described by Pérez-Elizalde et al. [52] and with the associated R function *margh.fun*. This method is based on estimation of the mode of the joint posterior distribution of $h$ and a form parameter $\varphi$. The shape and scale parameters of the gamma prior distribution for $h$ were 3.0 and 1.5, respectively.

The extensions of the GBLUP and RKHS models for multivariate analysis were used to predict the genomic estimated breeding values from data for the two sets of conditions. This approach is referred to hereafter as "multi-environment prediction". For both the extended GBLUP and RKHS models, the effects of markers are divided into two components: the main effect and the environment-specific effects [38,53]. Following the notation of Cuevas et al. [38], the model can be expressed as:

$$
\begin{bmatrix} y_1 \\ \vdots \\ y_j \\ \vdots \\ y_m \end{bmatrix} = \begin{bmatrix} 1\mu_1 \\ \vdots \\ 1\mu_j \\ \vdots \\ 1\mu_m \end{bmatrix} + \begin{bmatrix} X_1 \\ \vdots \\ X_j \\ \vdots \\ X_m \end{bmatrix} \beta_0 + \begin{bmatrix} X_1 & \dots & 0 & \dots & 0 \\ \vdots & \ddots & \vdots & \ddots & \vdots \\ 0 & \dots & X_j & \dots & 0 \\ \vdots & \ddots & \vdots & \ddots & \vdots \\ 0 & \dots & 0 & \dots & X_m \end{bmatrix} \begin{bmatrix} \beta_1 \\ \vdots \\ \beta_j \\ \vdots \\ \beta_m \end{bmatrix} + \begin{bmatrix} \varepsilon_1 \\ \vdots \\ \varepsilon_j \\ \vdots \\ \varepsilon_m \end{bmatrix}
$$

Where $y_j$ is the response vector in the $j$th environment, $\mu_j$ is the intercept in the $j$th environment, $X_j$ is the centered matrix of marker in the $j$th environment, $\beta_0$ is the vector of marker effects across all environments, $\beta_j$ is the vector of marker effects for environment $j$ and $\varepsilon_j$ is the random error for the $j$th environment. By using the following notation:

$$
y = \begin{bmatrix} y_1 \\ \vdots \\ y_j \\ \vdots \\ y_m \end{bmatrix}; \mu = \begin{bmatrix} 1\mu_1 \\ \vdots \\ 1\mu_j \\ \vdots \\ 1\mu_m \end{bmatrix}; u_0 = \begin{bmatrix} X_1 \\ \vdots \\ X_j \\ \vdots \\ X_m \end{bmatrix} \beta_0; u_E = \begin{bmatrix} X_1 & \dots & 0 & \dots & 0 \\ \vdots & \ddots & \vdots & \ddots & \vdots \\ 0 & \dots & X_j & \dots & 0 \\ \vdots & \ddots & \vdots & \ddots & \vdots \\ 0 & \dots & 0 & \dots & X_m \end{bmatrix} \begin{bmatrix} \beta_1 \\ \vdots \\ \beta_j \\ \vdots \\ \beta_m \end{bmatrix}; \varepsilon = \begin{bmatrix} \varepsilon_1 \\ \vdots \\ \varepsilon_j \\ \vdots \\ \varepsilon_m \end{bmatrix}
$$

the model can be written as follow:

$$y = \mu + u_o + u_E + \boldsymbol{\varepsilon}$$

In this mixed model, the random effect $u_o$ follows a multivariate normal distribution with a mean of zero and a variance–covariance matrix $K_0 \sigma_{u_o}^2$ with $\boldsymbol{K_0} = \boldsymbol{X_0 X_0'}/\boldsymbol{p}$ for the GBLUP and $\boldsymbol{K_0}(\boldsymbol{x_{0i}}, \boldsymbol{x_{0j}}) = \exp(-h_0 \| \boldsymbol{x_{0i}} - \boldsymbol{x_{0j}} \|^2)$ for RKHS. The random effect $u_E$ follows a multivariate normal distribution with a mean of zero and a variance–covariance matrix $K_E$ with

$$\boldsymbol{K_E} = \begin{bmatrix} \sigma_{u_1}^2 \boldsymbol{K_1} & \cdots & 0 & \cdots & 0 \\ \vdots & \ddots & \vdots & \ddots & \vdots \\ 0 & \cdots & \sigma_{u_j}^2 \boldsymbol{K_j} & \cdots & 0 \\ \vdots & \ddots & \vdots & \ddots & \vdots \\ 0 & \cdots & 0 & \cdots & \sigma_{u_1}^2 \boldsymbol{K_m} \end{bmatrix}$$

. The kernel matrix for each environment is estimate as follow: $\boldsymbol{K_j} = \boldsymbol{X_j X_j'}/\boldsymbol{p}$ for the GBLUP and $\boldsymbol{K_j}(\boldsymbol{x_{ji}}, \boldsymbol{x_{jj}}) = \exp(-h_j \| \boldsymbol{x_{ji}} - \boldsymbol{x_{jj}} \|^2)$ for RKHS

Analyses were performed in the R 3.6.1 environment [54]. The GBLUP and RKHS models were fitted in the R package *BGLR* 1.0.8 [55]. Inferences were based on 3,000 of the 35,000 iterations for the Gibbs sampler. The first 5,000 samples were discarded (burn-in) and we then kept one sample out of ten to avoid autocorrelation (thinning). Convergence of the Markov Chain-Monte Carlo algorithm was assessed for all parameters of the models, with Gelman-Rubin tests [56] and the R package *coda* 0.19–1 [57].

## Evaluation of predictive ability

**Cross-validation within the reference panel.** We estimated the predictive ability (PA) of the models described above using a cross-validation strategy within the RP: we randomly selected 80% of the panel to form the training set, with the remaining 20% used as the validation set. For multi-environment models, the genotypes composing the training set were associated with phenotypic information for the two sets of conditions, whereas no phenotypic information was available for those composing the validation set. This cross-validation approach is usually referred to as CV1 in the literature [58]. The random partitioning of the RP was repeated 100 times, and the PA for each partition was calculated as the Pearson coefficient of correlation between the genomic estimated breeding values and the corresponding phenotypes in the validation set. For each combination of model (single or multi-environment), statistical method (GBLUP, RKHS) and trait, the same partitions were used to calculate predictive ability. The resulting estimates of predictive ability were averaged, and the associated standard error was calculated. We analyzed the effect of the different factors (trait, conditions, prediction method, etc.) on PA, by performing analyses of variance. To avoid potential bias due to the distribution of the coefficient of correlation ($r \in [-1; 1]$), we transformed it using Fisher Z transformation according to the following equation:
$Z = 0.5\{\ln[1 + r] - \ln[1 - r]\}$. The analyses of variance were done on the Z statistics.

**Validation in the breeding population (BP).** We evaluated PA in the breeding population in two steps. We first used the 241 accessions of RP to train the model and then predicted the phenotype of each of the 393 lines in the BP. We then selected a set of 41 lines from the BP for actual phenotyping under the same conditions as the RP. The number of lines selected was chosen to allow for the phenotyping of three replicates in hydroponic conditions. This set of 41 lines (BP41) was chosen to be representative of the variability of the predicted phenotypic values of the most important salt tolerance traits observed for the 393 lines of the BP—Na, K

and Na/K—and for iSHOOT and iROOT, predicted by both the GBLUP and RKHS methods. We captured the variability present in the BP, for each trait, by selecting lines ranked in the lowest 10%, lines ranked in the top 10% and the lines with average performances in the following proportion: 20%, 20% and 60%. PA was calculated as the Pearson coefficient of correlation between the predicted phenotypes of the 41 lines obtained with models trained on the RP and the corresponding actual phenotypes.

## Results

### Characterization of the reference panel and the breeding population

In total, 20,255 informative SNPs spread throughout the genome were common to the RP and the BP (S1 Fig). The mean distance between SNPs was 18.3 kb, and the largest gap between markers was 1.3 Mb, on chromosome 11. Forty-two gaps of more than 500 kb were observed throughout the genome, with chromosomes 3 and 5 presenting the largest numbers of such gaps. SNPs in complete linkage disequilibrium ($r^2 = 1$) were removed. This resulted in 16,993 markers with a distribution very similar to the initial dataset (S1 Fig). The distribution of minor allele frequencies across non-redundant markers was similar for the RP and the BP (S2 Fig).

Analyses of the genetic structures of both the RP and BP highlighted the well-known bipolar structure of European rice accessions, with temperate and tropical *japonica* subgroups (S3 Fig). As expected, given the nature of the genetic material, the level of admixture between these two subgroups was high, reaching 37% in both the RP and the BP. The BP was highly related to the RP, as highlighted by the small genetic distance between the two populations (Fig 1). This relatedness between the two populations is a key parameter for genomic prediction.

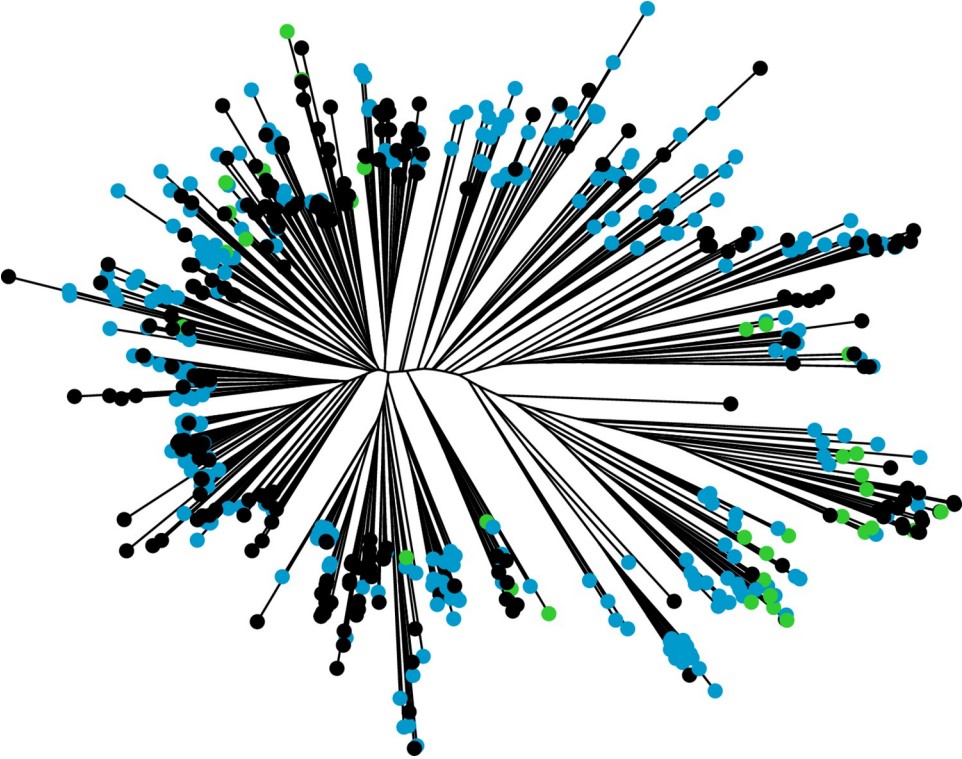

**Fig 1.** Unweighted neighbor-joining tree representing the dissimilarities between individuals composing the reference panel (black) and the breeding population (blue and green for the subset used for validation).

The 241 accessions of the RP had been evaluated under two hydroponic conditions: control and salt. Weak-to-moderate genotype x conditions interactions (G×C) were observed, depending on the trait (Fig 2). The weakest G×C interaction was that for leaf length (LL), with a rank correlation (Kendall) between the two conditions of $\rho = 0.79$. Conversely, specific leaf area (SLA) presented the strongest G×C interaction, with $\rho = 0.33$. The 10% of accessions with the

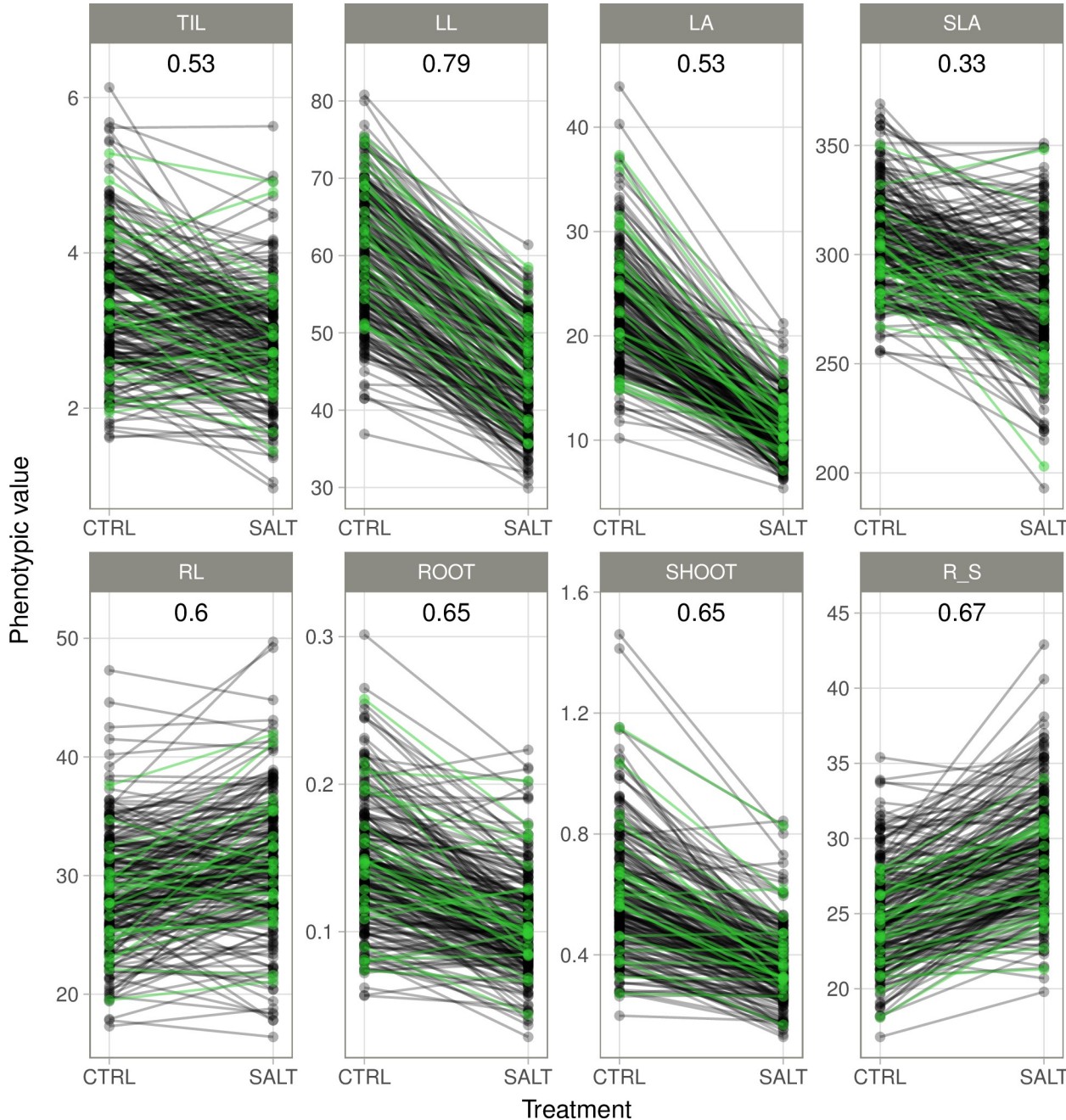

**Fig 2. Phenotypic performance of the accessions of the reference panel in the two sets of hydroponic conditions (CRTL: Control; SALT: Salt).** All traits measured in both conditions are presented: Number of tillers (TIL), leaf length (LL), leaf area (LA), specific leaf area (SLA), root length (RL), root dry weight (ROOT), shoot dry weight (SHOOT) and the ratio of root-to-shoot dry weights (R_S). The accessions in green are the 10% of accessions with the lowest Na/K ratios in salt conditions. Spearman's rank correlation coefficients ($\rho$) between control and salt conditions are indicated at the top of each panel.

lowest Na/K ratios did not differ significantly (*p*-value > 0.05) from the rest of the RP for any combination of trait/conditions other than root/shoot ratio (R_S) in salt conditions, for which the ratio was higher for the rest of the panel. The correlation between morphological traits ranged from -0.45 to 0.90 in control conditions and from -0.37 to 0.90 in salt-stress conditions (S2 Table). ROOT was strongly correlated with SHOOT (0.90) and LL was significantly correlated with ROOT, SHOOT and LA, in both sets of conditions. However, the correlations between traits tended to be weaker in salt conditions than in control conditions. In salt conditions, the ion mass fractions presented few significant correlations with morphological traits: NA/K was correlated with TIL (-0.20), SHOOT (-0.19) and R_S (0.22, S2 Table).

The phenotypic variability observed in the RP was partly related to population structure. Indeed, Na/K was significantly higher for the admixed subgroup than for the tropical and temperate subgroups (S4 Fig). A difference was also found for RL, for which the admixed subgroup had lower values, suggesting a higher susceptibility to the effects of salinity on root development (S4 Fig).

## Predictive ability of genomic prediction within the reference panel

**Comparison between single and multi-environment models.** Cross-validation in the RP gave predictive ability (PA) values for the multi-environment model ranging from 0.25 to 0.64 in control conditions and from 0.38 to 0.63 in salt conditions, for the eight morphological traits analyzed (Fig 3). Similar PAs were obtained for the eight traits in the single environment

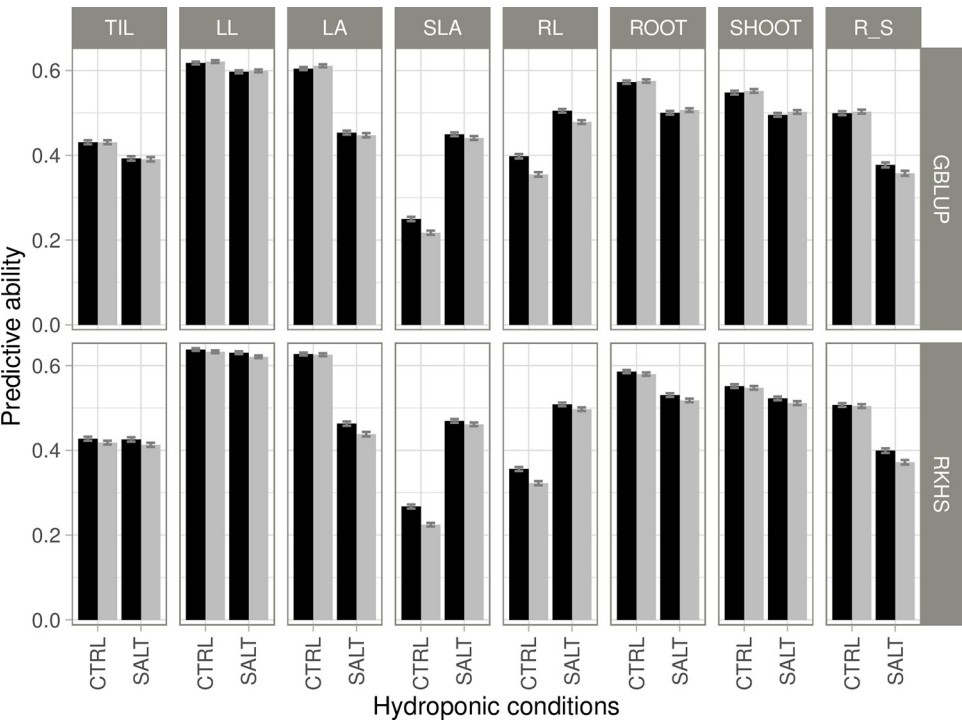

**Fig 3. Estimates of predictive ability for performances in the two sets of conditions (CTRL and SALT) for the reference panel.** Performances were predicted with multi-environment (black) or single-environment (gray) models. Two different prediction methods were used: GBLUP and RKHS. The traits considered were: Number of tillers (TIL), leaf length (LL), leaf area (LA), specific leaf area (SLA), root length (RL), root dry weight (ROOT), shoot dry weight (SHOOT) and the ratio of root-to-shoot dry weights (R_S). The bars represent the average predictive ability over 100 replicates, and the error bars represent the standard error of the mean.

model, with values ranging from 0.22 to 0.63 in control conditions and from 0.36 to 0.62 in salt conditions. For both the single-environment and multi-environment models, the lowest PA was that for SLA and the highest was that for LL in control conditions. PA was higher in control conditions than in salt conditions for six of the eight traits (TIL, LL, LA, SHOOT, ROOT, and R_S) (Fig 3). The largest positive difference in accuracy (0.19) between control and salt conditions was for LA when the single environment model was used. For SLA and RL, PA was higher under salt conditions than under control conditions, with differences up to 0.24 and 0.17 (single-environment model, RKHS), respectively. All the factors considered in the analysis of variance (trait, conditions, prediction method and prediction model) had a significant effect on PA, with the largest effect for trait, the other factors being of a much lesser importance (S3 Table). Indeed, differences in PA between single and multi-environment models were close to zero, except for SLA and RL in control conditions for which accuracy was 15% and 10% higher, respectively, for the multi-environment model than for the single-environment model. Interestingly, these were the two traits with the best PA in salt conditions. No clear relationship was found between the strength of G×C interactions for traits and the performance of the multi-environment model relative to that of the single-environment model.

**Stress response indices and ion mass fractions.** The stress response indices combined values from both control and salt conditions, whereas mass fractions were measured only under salt conditions. Only the single environment model could, therefore, be used for mass fractions. The PA values for the indices were lower than those of other traits in each set of conditions, with values ranging from -0.05 to 0.35 (Fig 4A). On average, over the two prediction methods, the iROOT, iSHOOT and iLL indices had the lowest PAs (lower than 0.10). Conversely, the PAs for iRL and iSLA were greater than 0.30. The PA for Na and K mass fractions and for Na/K ranged from 0.20 to 0.40 (Fig 4A). The K mass fraction was slightly better predicted than Na mass fraction, with PAs of 0.34 (GBLUP) and 0.40 (RKHS), versus 0.28 (GBLUP) and 0.29 (RKHS), respectively. Both trait and prediction method had a significant effect on PA. The effect of trait was the most significant, with prediction method having only a marginal effect (S4 Table). A negative correlation was found between PA and the strength of G×C interactions for the trait: -0.69 (*p*-value = 0.059) and -0.58 (*p*-value = 0.134) for GBLUP and RKHS, respectively (Fig 4B).

## Validation of predictive ability in the breeding population

The performances of the 393 genotypes of the BP were predicted with the two prediction models (single and multi-environment) built on the RP. As expected, the genomic estimated performances were shrunk toward the mean value of the RP, and this effect was more pronounced for RKHS than for GBLUP. Depending on the trait, the coefficients of correlation (ρ) between the predicted performances estimated with RKHS and GBLUP ranged from 0.88 (SLA) to 0.98 (LL) for the multi-environment model and from 0.76 (SLA) to 0.99 (LL) for the single-environment model (S5 Table). For indices and ion mass fractions, the coefficients of correlation ranged from 0.29 (iROOT) to 0.97 (iLA). The traits presenting the lowest correlation between prediction methods were those with the lowest PAs in cross-validation in the RP: iROOT, iSHOOT, and iLL. For Na and K, and for Na/K, the coefficients of correlation between prediction methods exceeded 0.90.

For validation of the predicted performances, a subset of 41 lines from the BP (BP41) was selected for phenotyping. This selection had little effect on the extent of variability of the predicted traits relative to the entire BP (S5 and S6 Figs), but it decreased the neutral genetic variability, as shown by the distribution of BP41 on the neighbor-joining tree (Fig 1). The repeatability of the actual phenotypic data obtained for BP41 through evaluation under control

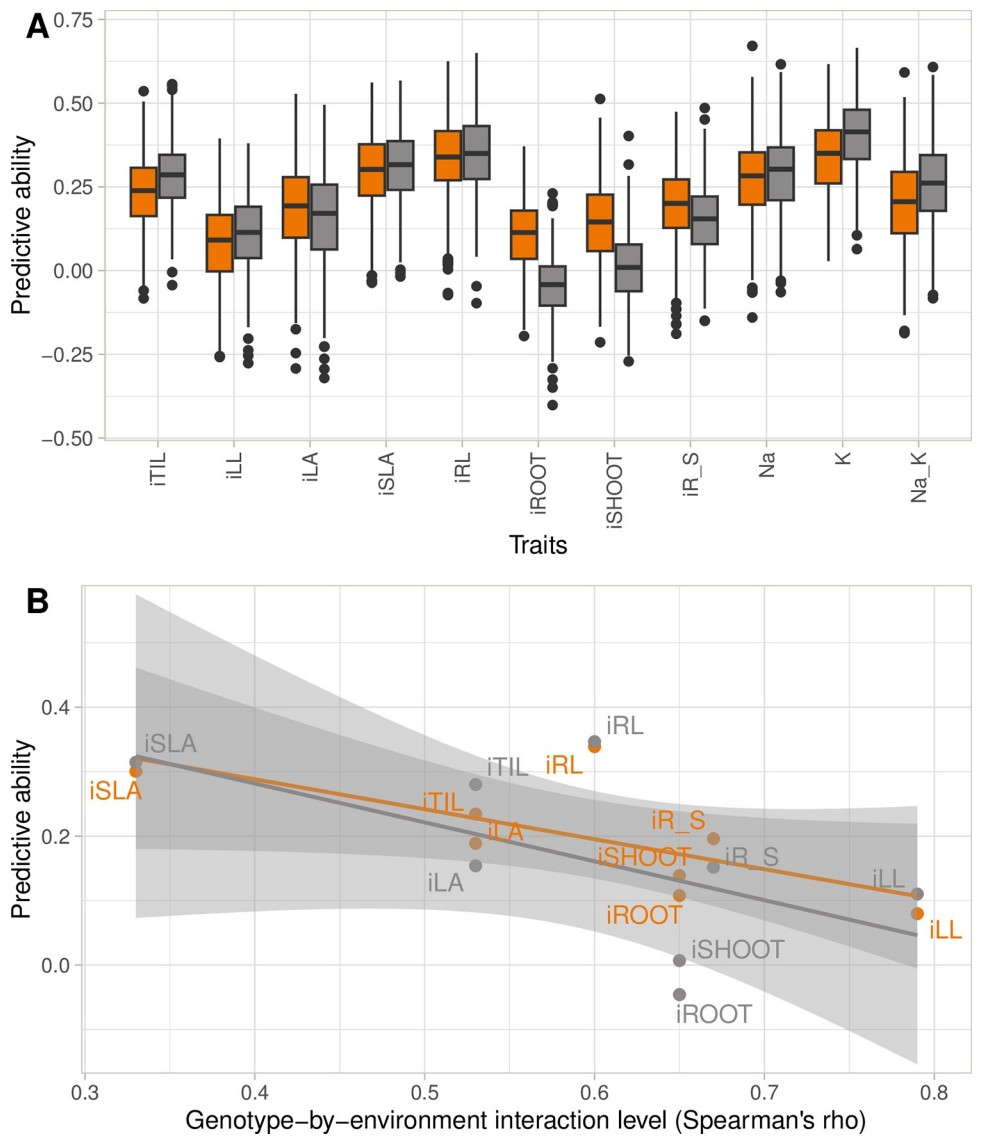

**Fig 4.** (A) Predictive abilities for stress response indices (iTrait) and ion mass fraction (Na and K) in the reference panel. Two different prediction methods are presented: GBLUP in orange and RKHS in gray. A Tukey boxplot is used to represent the data. (B) Scatterplot of estimates of predictive ability for indices and the level of genotype-by-environment interactions estimated by calculating the Spearman correlation coefficient for the same genotype in control and salt conditions.

and salt conditions ranged from 0.70 to 0.93 for the measured traits and from 0.44 to 0.55 for the indices (Table 1). Under salt conditions, the selected lines displayed phenotypic variation for ion mass fraction (Na and K) within the expected range relative to the susceptible and tolerant controls (Fig 5). G×C interactions of similar strength (ρ) to those in the RP were found (Table 1). The weakest G×C interaction was that for LA, with a rank correlation ρ between the two conditions of 0.83, and the strongest G×C interaction was that for SLA (ρ = 0.25).

The phenotypic data for BP41 were used to estimate the PA of the models trained on the RP for the various traits. For the multi-environment model, PA ranged from 0.00 to 0.66 and from 0.09 to 0.54 in control and salt conditions, respectively (Table 1). As for cross-validation, similar results were obtained with the single-environment model, with values ranging from

**Table 1. Summary of statistics for the 41 lines of the breeding population (BP41) for the various traits measured under control and salt conditions.**

| | Prediction method | TIL | LL | LA | SLA | RL | ROOT | SHOOT | R_S | Na* | K* | Na/K* |
|---|---|---|---|---|---|---|---|---|---|---|---|---|
| $H^2$ (control/salt) | - | 0.70 / 0.73 | 0.89 / 0.93 | 0.88/ 0.87 | 0.48/ 0.07 | 0.88 / 0.69 | 0.82 / 0.86 | 0.89 / 0.89 | 0.87 / 0.76 | 0.82 | 0.85 | 0.82 |
| ρ | - | 0.38 | 0.66 | 0.83 | 0.25 | 0.67 | 0.31 | 0.54 | 0.80 | - | - | - |
| PA with multi-environment model (control/salt) | RKHS | 0.02 / 0.14 | 0.38 / 0.56 | 0.36 / 0.50 | 0.50 / 0.14 | 0.63 / 0.45 | 0.66 / 0.4 | 0.43 / 0.43 | 0.48 / 0.31 | - | - | - |
| | GBLUP | 0.00 / 0.10 | 0.43 / 0.54 | 0.36 / 0.45 | 0.57 / 0.09 | 0.64 / 0.36 | 0.65 / 0.29 | 0.42 / 0.34 | 0.47 / 0.41 | - | - | - |
| PA with single-environment model (control/salt) | RKHS | 0.03 / 0.14 | 0.37 / 0.59 | 0.37 / 0.53 | 0.45 / 0.17 | 0.61 / 0.42 | 0.66 / 0.45 | 0.46 / 0.47 | 0.48 / 0.28 | - | - | - |
| | GBLUP | 0.02 / 0.04 | 0.39 / 0.61 | 0.37 / 0.49 | 0.56 / 0.12 | 0.64 / 0.32 | 0.67 / 0.36 | 0.46 / 0.40 | 0.41 / 0.40 | - | - | - |
| PA for the indices and ion content | RKHS | -0.05 | 0.21 | 0.11 | 0.35 | 0.19 | 0.33 | 0.34 | 0.02 | 0.31 | 0.26 | 0.26 |
| | GBLUP | 0.01 | 0.06 | 0.13 | 0.32 | 0.31 | 0.26 | 0.33 | -0.07 | 0.32 | 0.35 | 0.25 |

Repeatability in the two conditions ($H^2$), the rank correlation between conditions (ρ),predictive abilities (PA) for single- and multi-environment models and for the indices are provided. The traits presented are: Number of tillers (TIL), leaf length (LL), leaf area (LA), specific leaf area (SLA), root length (RL), root dry weight (ROOT), shoot dry weight (SHOOT), the ratio of root-to-shoot dry weights (R_S), the ion mass fractions of Na and K and their ratio (Na/K). * The ion mass fractions were measured only in salt conditions.

0.02 to 0.67 in control conditions and from 0.04 to 0.61 in salt conditions. For stress-response indices and ion mass fractions, PA ranged from -0.07 to 0.35. The PAs for the Na (0.31 and 0.32, with RKHS and GBLUP, respectively) and K (0.26 and 0.35) mass fractions (the main traits used to select BP41) were intermediate, lying between those for the other traits. The differences between RKHS and GBLUP were slightly larger than those observed for cross-validation. Depending on the trait and the model, gains in PA of up to 0.10 were observed for RKHS (SHOOT in salt conditions) or for GBLUP (SLA in control conditions). Interestingly, the PAs estimated by cross-validation and those obtained with the subset of the BP were not correlated for control conditions, whereas there was a non-significant trend towards a positive correlation between these PAs in salt conditions (Fig 6, S6 Table).

## Discussion

### Performance of *japonica* accessions under salt stress

In this study, we evaluated the salt tolerance of accessions and advanced lines from European breeding programs at the seedling stage under hydroponic conditions, with a no-salt control and a salt-stress treatment of 50 mM NaCl (6.5 dS·m$^{-1}$). Hydroponic experiments are very useful for evaluations of the sensitivity of accessions to a given level of salt stress [6,59]. This approach has been used extensively to screen tolerant material in breeding programs [11] as salt levels are highly variable in field experiments, with micro-environmental and seasonal variations that can bias the evaluation of accessions. However, field-based evaluations and screening in controlled conditions are jointly used in breeding programs as stress tolerance at the seedling stage may not fully correlate with field performance. In our experiments, a moderate stress level, corresponding to a degree of salinity commonly observed in the Camargue region of France was used [60]. Considerable variability was observed in the phenotypic response to salt stress in both the RP and BP41, consistent with the findings of previous studies on *japonica* accessions [15,16,61]. In their work on 176 temperate *japonica* accessions, Batayeva et al. [61] reported that only a few accessions (Nep Ngau, Bai Mang Ai Zhong, and Shinchiku Iku 97)

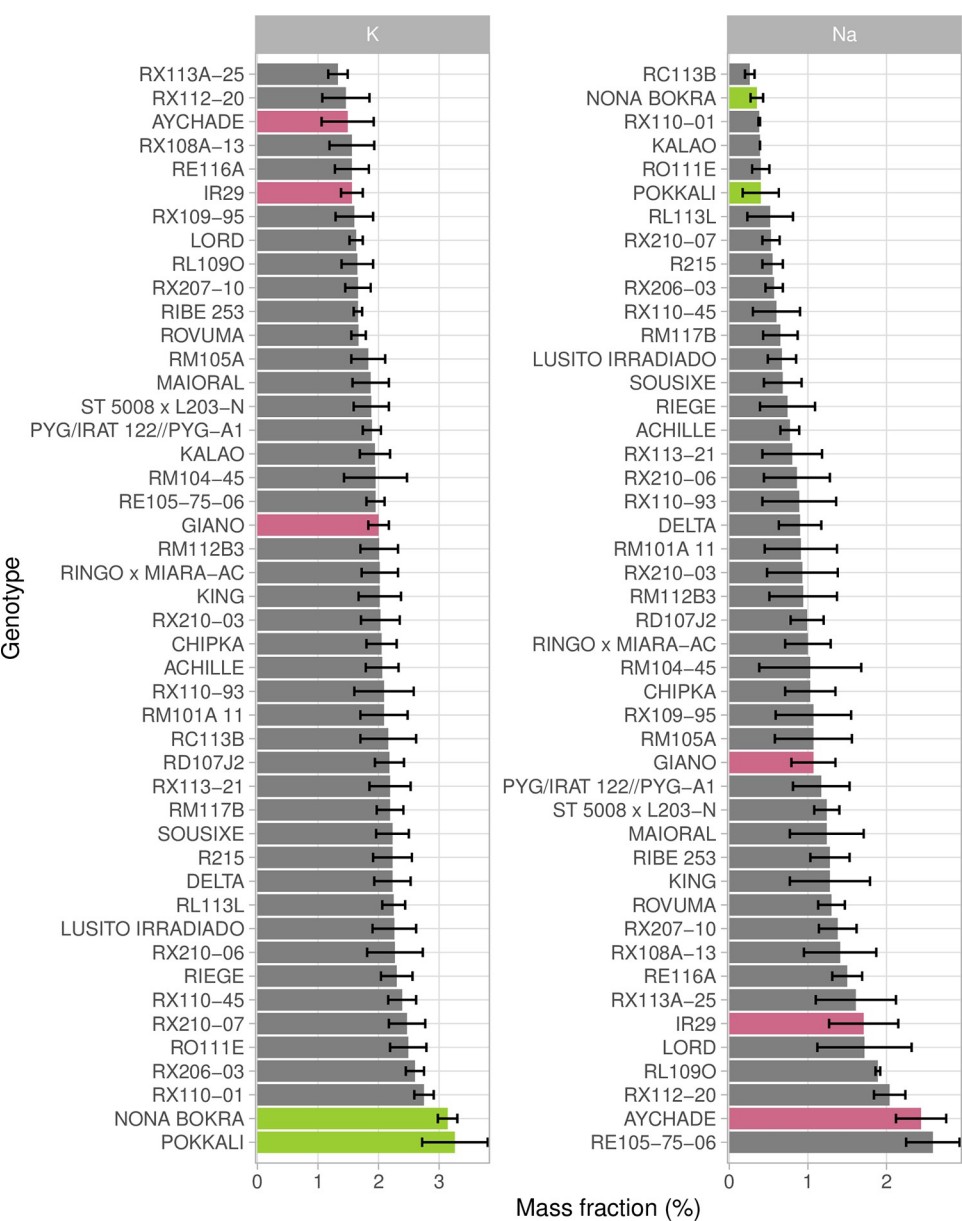

**Fig 5. Distribution of the ion mass fractions for Na and K in the subset of the breeding population (BP41).** The two tolerant controls (Nona Bokra and Pokkali) are represented in green and the three susceptible controls (IR29, Giano and Aychade) are shown in pink. The lines from BP41 are in gray. The error bars represent the confidence interval of the mean (95%) which was calculated based on the data for the three replicates for each line.

were as tolerant as the control variety, the others being moderately tolerant or susceptible. In this study, we found that three lines had performances similar to that of the tolerant control, with low Na_K values under salt conditions (Fulgente, Escarlate and RX110_01).

## Accuracy of genomic predictions for salt tolerance

Genomic prediction has been studied in detail in rice in recent years, with more than 50 studies published to date [33]. However, none of these studies focused on predicting genotype performances under salt stress. Depending on the trait, the type of population, the validation

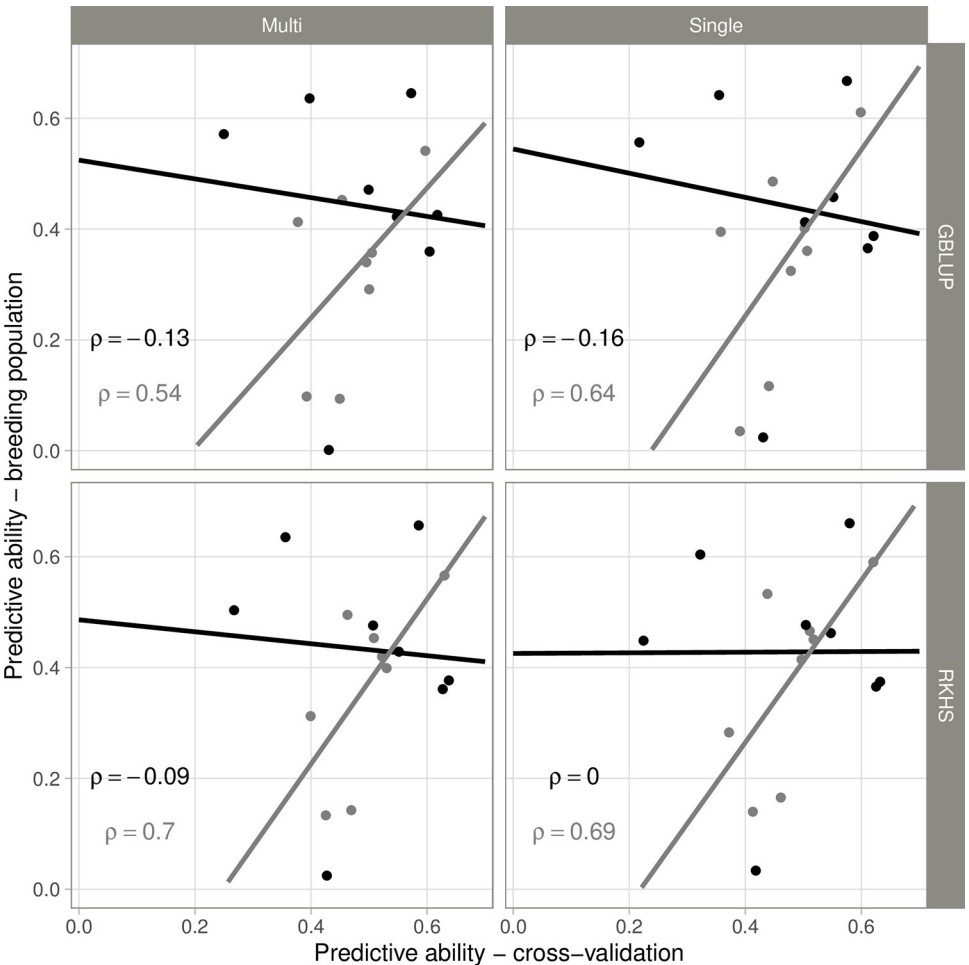

**Fig 6. Comparison of the predictive abilities for eight morphological traits estimated by cross-validation on the reference panel and the 41 selected lines from the breeding population used as a validation set.** Control conditions are shown in black, and salt conditions in gray. Spearman's rank correlation coefficients (ρ) are indicated.

method and the prediction method, the accuracies reported in these studies varied substantially, from close to zero to more than 0.90. Here, prediction accuracy ranged from 0.05 to 0.63 for the various traits when the cross-validation method was applied to the RP. No clear differences were found between the two prediction methods used (GBLUP and RKHS). These results are consistent with those reported in previous studies [34,51,62,63]. We found that the differences between single- and multi-environment models were also small, although multi-environment models outperformed single-environment models for most of the traits and tended to be more accurate for predictions in salt conditions. The small differences between single- and multi-environment models were expected, given the type of cross-validation scheme used, with the prediction of untested genotypes only. With this type of cross-validation scheme, multi-environment models and single-environment models tend to perform similarly [34,64,65]. Multi-environment models have been shown to perform better when the predicted genotypes are evaluated in at least one environment (mimicking sparse testing evaluation). In this case, the gain in prediction accuracy of multi-environment models over single-environment models may be between 30% and 50%, as shown by Ben Hassen et al. [34] with phenotypic data for two sets of conditions (continuous flooding and alternate wetting and drying).

Cuevas et al. [38] reported similar results for a wheat dataset including four environments, with gains of up to 60%, depending on the traits considered and the validation method.

Stress response indices had lower predictive abilities than regular traits. For the indices, predictive ability was negatively correlated with the strength of G×C interactions. This result reflects the difficulty in predicting and selecting for relative performance between a stress environment (in our case salt stress) and normal conditions. The positive relationship between predictive ability and heritability is well known and has been described in studies on genomic prediction [66,67], but the impact of genotype-by-environment interactions on predictive ability tends to be more difficult to characterize [68,69]. Furthermore, for indices, the errors associated with measurements in the two sets of conditions are propagated in the index, tending to decrease heritability. Ion mass fractions (Na and K) and their ratio (Na/K) had low-to-moderate predictive abilities (0.2 to 0.4) with similar values in both populations (RP and BP41). These predictive abilities are in the expected range given the complex genetic architecture underlying these two traits [41]. No other study on rice has assessed the performance of genomic prediction in the context of salinity tolerance. We were therefore unable to make direct comparisons for this trait. However, the predictive abilities obtained in this study for Na and K could potentially be improved with prediction methods modeling marker effects differently from GBLUP and RKHS. Methods such as Bayesian LASSSO, BayesB and Random Forest have been shown to perform better when only a few regions of the genome have moderate effect on the target traits [51,70,71]. Similarly, the integration of GWAS results into the genomic prediction model might also increase prediction accuracy [35,72,73]. In this study, no major QTL was found in the RP [41], but this approach may be of interest when major QTLs, such as Saltol are segregating in the breeding population.

## Importance of validation on selection candidates

Predictive ability or accuracy is routinely used to evaluate the efficiency of genomic prediction models [74]. In most genomic selection studies in plants, accuracy is measured as the correlation (r) between observed and predicted phenotypic performances [75]. The most common approach for estimating accuracy is cross-validation (subset validation), because of its ease of use. In this approach, the dataset is split in two (the calibration set and the validation set), making it possible to estimate accuracies in a given population while keeping parameters such as marker density, population structure or allele frequencies constant. However, cross-validation tends to overestimate accuracy relative to other validation approaches, such as inter-set validation or progeny validation [63,76,77]. Most of the studies performed in rice have used cross-validation to obtain estimates of prediction accuracy [33]. Here, we used both cross-validation and inter-set validation (validation in the breeding population). The two approaches gave similar estimates of predictive ability. The accuracies obtained by cross-validation and with BP41 were well-correlated, but only in salt conditions. This finding reflects the method used to select lines for inclusion in BP41: ion mass fractions measured under salt conditions. The use of similar phenotyping conditions for model training and validation is therefore important, to obtain a more precise idea of the level of accuracy that can be expected, as GxE interactions generally decrease accuracy [69,78]. The relatedness between the RP and the BP, with a similar structure in the two populations, may also explain this result. Indeed, the genetic distance between the training set and the validation set has been shown to be one of the major factors affecting accuracy [79,80]. BP41 constitutes only a small subset of the entire BP, but most of the parental accessions and closely related lines, were present in the RP used to train the model. Ben Hassen et al. [63] reported similar results in a study using a diversity panel to predicted advance material from the breeding program.

## Implications for breeding for salt tolerance in rice

Genotypes tolerant to salt stress are clearly less common among *japonica* accessions than among *indica* accessions [15,16]. However, it is important for temperate rice breeding programs to characterize their material better for salinity tolerance because the sources of tolerance identified in *indica* accessions are difficult to introgress due to the complex genetic architecture of the trait, as revealed by linkage mapping [81] and association studies [16,41,82]. Sterility problems have also been reported for intersubspecific *indica* x *japonica* crosses [83,84]. We show here that genomic selection can be used to predict the salt tolerance of material from European rice breeding programs (*japonica*) under mild constraints. The major challenge would be efficiently combining evaluations under normal and salt conditions. As discussed above, capturing GxE interactions (or GxC interactions) with multi-environment models can increase the accuracy of predictions. Genomic selection for salt tolerance can be implemented just after line fixation (usually F6). In breeding programs involving rapid generation advances, such selection can take place in the third year of the breeding strategy [33]. The use of genomic selection at this stage would make it possible to select lines on the basis of their predicted performance in both normal and salt conditions. Multi-environment genomic prediction could be used to combine information from normal and salt conditions (either in the greenhouse or in the field) for prediction in a larger set of candidates from the cohort of the next cycle. The selection intensity for salt tolerance would be increased early in the breeding scheme (e.g., stage 1 yield trial). As stress trials are difficult to manage, a targeted set of lines could be evaluated in stress conditions, but with a higher degree of replication, to update the model.

## Supporting information

**S1 Fig.** Distribution in the rice genome of the informative markers for the complete set of 20,255 SNPs (upper panel) and the non-redundant set of 16,993 SNPs (lower panel). (PDF)

**S2 Fig. Distribution of minor allele frequency (MAF) for the 16,993 non-redundant SNPs in the two populations: The reference panel and the breeding population.** (PDF)

**S3 Fig. Unweighted neighbor-joining tree and the associated genetic structure estimated for $K = 2$ in the upper panel and $K = 3$ in the lower panel.** Temperate *japonica* is shown in red, tropical *japonica* in blue and admixed accessions are shown in purple. (PDF)

**S4 Fig. Boxplot for the stress response indices (iTrait) and the K and Na mass fractions and their ratio in the reference panel.** The different subpopulations (admixed, temperate, tropical) were defined with molecular markers (see materials and methods). Different letters for a given trait indicate a significant difference between group means (Tukey's HSD test, $p < 0.05$). (PDF)

**S5 Fig. Boxplot of genomic estimated breeding value (GEBV) for the eight morphological traits in the breeding population of 393 lines.** The 41 lines selected for the validation experiment are represented in black and the rest of the population is shown in gray. Two prediction methods (GBLUP and RKHS) and two models (single- and multi-environment) were compared. (PDF)

**S6 Fig. Distribution of genomic estimated breeding value (GEBV) for Na and K mass fraction and their ratio (Na/K) in the breeding population of 393 lines.** The 41 breeding lines selected for the validation experiment are represented in green and the rest of the population is shown in gray. Two different prediction methods were used: GBLUP and RKHS.
(PDF)

**S7 Fig. Assessment of salinity tolerance under hydroponic conditions.** The image at the top represents half the 12 tanks for one replicate for the reference panel. The image at the bottom represents the three replicates for the selected genotypes of the breeding population. The control tanks are shown on the left and those for salt conditions are shown on the right.
(PDF)

**S1 Table. List of accessions in the two populations: The reference panel and the breeding population.**
(XLSX)

**S2 Table. Correlations between morphological traits in control conditions (upper table) and between ion mass fractions in salt conditions (lower table).** The Spearman rank correlation coefficients are displayed in the lower part of the matrices and the associated *p*-value are shown in the upper part.
(PDF)

**S3 Table. Analysis of variance of predictive abilities in the reference panel for performances in both sets of conditions (CTRL and SALT).** Two prediction methods were compared (GBLUP and RKHS), for eight traits and two models (single- and multi-environment).
(PDF)

**S4 Table. Analysis of variance of predictive abilities in the reference panel for indices and ion mass fractions (referred to as Trait in the table).** Two prediction methods were compared (GBLUP and RKHS).
(PDF)

**S5 Table. Spearman's rank correlation coefficient for the relationship between the predicted performances estimated with RKHS and GBLUP for the entire breeding population, with single- and multi-environment models.**
(PDF)

**S6 Table. Relationship between predictive abilities estimated by cross-validation on the reference panel and those estimated with the subset (41 lines) of the breeding population.** Two models (single- and multi-environment) and two methods (GBLUP and RKHS) were evaluated.
(PDF)

## Acknowledgments

This work has been realized with the support of MESO@LR-Platform at the University of Montpellier. The authors thank US49 from CIRAD for conducting the mass fraction analyses.

## Author Contributions

**Conceptualization:** Nourollah Ahmadi, Brigitte Courtois.

**Data curation:** Jérôme Bartholomé, Julien Frouin, Laurent Brottier, Tuong-Vi Cao, Arnaud Boisnard, Brigitte Courtois.

**Formal analysis:** Jérôme Bartholomé, Julien Frouin, Brigitte Courtois.

**Funding acquisition:** Nourollah Ahmadi, Brigitte Courtois.

**Investigation:** Jérôme Bartholomé.

**Methodology:** Jérôme Bartholomé, Brigitte Courtois.

**Project administration:** Nourollah Ahmadi, Brigitte Courtois.

**Supervision:** Brigitte Courtois.

**Visualization:** Jérôme Bartholomé.

**Writing – original draft:** Jérôme Bartholomé.

**Writing – review & editing:** Julien Frouin, Laurent Brottier, Tuong-Vi Cao, Arnaud Boisnard, Nourollah Ahmadi, Brigitte Courtois.

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
