## [Decision Letter · Decision Letter 0]

4 Jul 2023

PONE-D-23-12298Genomic selection for salinity tolerance in japonica ricePLOS ONE

Dear Dr. Bartholomé,

Thank you for submitting your manuscript to PLOS ONE. After careful consideration, we feel that it has merit but does not fully meet PLOS ONE’s publication criteria as it currently stands. Therefore, we invite you to submit a revised version of the manuscript that addresses the points raised during the review process.

We look forward to receiving your revised manuscript.

Kind regards,

Muhammad Abdul Rehman Rashid, PhD

Academic Editor

PLOS ONE

Journal Requirements:

Reviewers' comments:

Reviewer's Responses to Questions

**Comments to the Author**

1. Is the manuscript technically sound, and do the data support the conclusions?

Reviewer #1: Yes

Reviewer #2: Yes

2. Has the statistical analysis been performed appropriately and rigorously? 

Reviewer #1: Yes

Reviewer #2: Yes

3. Have the authors made all data underlying the findings in their manuscript fully available?

Reviewer #1: Yes

Reviewer #2: Yes

4. Is the manuscript presented in an intelligible fashion and written in standard English?

Reviewer #1: Yes

Reviewer #2: Yes

5. Review Comments to the Author

Reviewer #1: The authors of this study carried out a genomic prediction to predict a series of morphological and ion mass fraction traits in rice. For model development, they used a large reference panel of 241 japonicas and presented cross-validation results. They used the model to predict all traits in a separate breeding panel of 339 lines and went the extra step to phenotype 41 of those lines. The 41 phenotyped BP lines were selected to represent the variation observed in the predicted phenotypes in several traits. Overall, I felt this was a very well-designed and well-described study. I have no major issues with it. Please find some minor comments below that I hope can help improve the manuscript.

Specific comments

I noticed there are a lot of bar charts in this paper. The authors may want to consider replacing bar charts with boxplots, which are more informative for the distribution of accuracies presented. Bar charts tend to hide a lot of detail. Also, the figure legends for the bar charts don’t explain what the bars and error bars actually represent (mean and standard error of the mean, I’m guessing, but it should be stated explicitly).

In the Discussion section ‘implications for breeding for salt tolerance in rice’ I wonder why the authors do not discuss whether evaluation in hydroponics for salt tolerance extends to conditions experienced in the field. For example, do breeders for rice under saline conditions currently use hydroponics for their screenings? I think this is an important discussion point to make.

I may have missed this in the Methods, but can the authors please clarify what their multi-environments are? Since this is a hydroponic study, are the environments the different replicates in time or something else?

Reviewer #2: The MS PONE-D-23-12298 by Bartholomé et al. entitled "Genomic selection for salinity tolerance in japonica rice" reports on data from an investigation aimed at assessing the accuracy of genomic prediction models, considering genotype x environment interactions, for performance related to tolerance to mild salt stress in japonica rice accessions at the seedling stage and validate these models. Several reasons make the investigation quite innovative; among them are, other than the almost total absence of such predicting models for salt tolerance in rice, the focus on japonica rice accessions (less characterized for what concern salt tolerance and related breeding programs in comparison to indica accessions) and the exposition of the accessions to a mild salt condition (6.5 dS m-1) that is more probable occur in the fields than those more severe (8.0-10.0 dS m-1) too often considered in the studies carried out in controlled environments.

The experimental and data elaboration phases were well planned, carried out, described, and discussed. The logical cascade adopted in the text allows the reader to follow the MS clearly. The conclusions are well sustained by the results obtained. The predictive models proposed actually seem to be promising. Therefore, I think the MS is suitable for

publication with just some minor revisions and introducing some brief comments, as I suggest below.

-The authors should justify why they did not adopt the Stress Susceptibility Index (SSI; see Fischer and Maurer, 19878, Aust. J. Agr. Res. 29,897–912. doi: 10.1071/AR9780897) rather than iTRAIT as the stress index. Indeed, the former expresses better the behaviour of each accession in response to the stress with respect to the mean behaviour the entire panel considered. I welcome comments by the authors about this point, even without introducing it in the text.

-It is unclear why the authors excluded the possibility of evaluating the Na+ and K+ mass fraction of shoot tissue in the plants grown under the control condition (page 16, lines 354-355). In the control nutrient solution adopted (page 8, line 173; reference #41), the Na+/K+ molar ratio is about 0.18; thus, the Na+ and K+ mass fraction of shoot tissue could be detected. If the ion levels were below (unlikely) the detection limit of the ICP-AES technique adopted, an ICP-MS approach might be applied. Could the authors exclude that also for the NA/K trait the stress index had been used different results would have been obtained? Could the authors comment on this aspect?

-Finally, other very minor points:

· What do the error bars in Fig.3 and 4 indicate? Please, introduce this information in the figure legends.

· Why the Na+/K+ ratios were evaluated from starting on ions mass faction and not as molar ratios?

6. PLOS authors have the option to publish the peer review history of their article (what does this mean?). If published, this will include your full peer review and any attached files.

Reviewer #1: No

Reviewer #2: No

---

## [Author Response · Author response to Decision Letter 0]

31 Aug 2023

Response to Reviewers

Reviewer #1: The authors of this study carried out a genomic prediction to predict a series of morphological and ion mass fraction traits in rice. For model development, they used a large reference panel of 241 japonicas and presented cross-validation results. They used the model to predict all traits in a separate breeding panel of 339 lines and went the extra step to phenotype 41 of those lines. The 41 phenotyped BP lines were selected to represent the variation observed in the predicted phenotypes in several traits. Overall, I felt this was a very well-designed and well-described study. I have no major issues with it. Please find some minor comments below that I hope can help improve the manuscript.

Thank you for this positive evaluation of our study. 

Specific comments

I noticed there are a lot of bar charts in this paper. The authors may want to consider replacing bar charts with boxplots, which are more informative for the distribution of accuracies presented. Bar charts tend to hide a lot of detail. Also, the figure legends for the bar charts don't explain what the bars and error bars actually represent (mean and standard error of the mean, I'm guessing, but it should be stated explicitly).

We hadn't necessarily paid attention to it, but there are, indeed, three figures with bar charts. We wanted to focus on the average predictive abilities, but this can mask the variability associated with repetitions. We changed panel A of Fig 4 with boxplots. For Figs 3 and 5, we preferred to keep the bar charts. For Fig 3, the boxplots made the figure less easy to read with too much information. For Fig 5, boxplots are not suitable since we have only three data points per genotype. For these two figures, we clarified what represent the bars and the error bars.

In the Discussion section' implications for breeding for salt tolerance in rice' I wonder why the authors do not discuss whether evaluation in hydroponics for salt tolerance extends to conditions experienced in the field. For example, do breeders for rice under saline conditions currently use hydroponics for their screenings? I think this is an important discussion point to make.

This idea was partly presented in the first paragraph of the discussion (Hydroponic experiments are very useful for evaluations of the sensitivity of accessions to a given level of salt stress (6, 59). Salt levels are highly variable in field experiments, with micro-environmental and seasonal variations that can bias the evaluation of accessions). In breeding programs, the two approaches are used to select tolerant lines. Selection at seedling stage in controlled conditions is usually done to screen a large number of genotypes early in the breeding scheme and later field evaluations (both stress and non-stress) are conducted on a smaller number of selection candidates. 

We modified the following paragraph of the discussion to clarify this point : This approach has been used extensively to screen tolerant material in breeding programs (11) as salt levels are highly variable in field experiments, with micro-environmental and seasonal variations that can bias the evaluation of accessions. However, field-based evaluations and screening in controlled conditions are jointly used in breeding programs as stress tolerance at the seedling stage may not fully correlate with field performance.

I may have missed this in the Methods, but can the authors please clarify what their multi-environments are? Since this is a hydroponic study, are the environments the different replicates in time or something else?

We are using the two conditions (control and salt) as the two "environments" for the multi-environment models. This was formulated as follows in the manuscript: The extensions of the GBLUP and RKHS models for multivariate analysis were used to predict the genomic estimated breeding values from data for the two sets of conditions. This approach is referred to hereafter as "multi-environment prediction".

Reviewer #2: The MS PONE-D-23-12298 by Bartholomé et al. entitled "Genomic selection for salinity tolerance in japonica rice" reports on data from an investigation aimed at assessing the accuracy of genomic prediction models, considering genotype x environment interactions, for performance related to tolerance to mild salt stress in japonica rice accessions at the seedling stage and validate these models. Several reasons make the investigation quite innovative; among them are, other than the almost total absence of such predicting models for salt tolerance in rice, the focus on japonica rice accessions (less characterized for what concern salt tolerance and related breeding programs in comparison to indica accessions) and the exposition of the accessions to a mild salt condition (6.5 dS m-1) that is more probable occur in the fields than those more severe (8.0-10.0 dS m-1) too often considered in the studies carried out in controlled environments.

The experimental and data elaboration phases were well planned, carried out, described, and discussed. The logical cascade adopted in the text allows the reader to follow the MS clearly. The conclusions are well sustained by the results obtained. The predictive models proposed actually seem to be promising. Therefore, I think the MS is suitable for publication with just some minor revisions and introducing some brief comments, as I suggest below.

We thank the Reviewer for carefully reviewing the manuscript and his interest in this study.

-The authors should justify why they did not adopt the Stress Susceptibility Index (SSI; see Fischer and Maurer, 19878, Aust. J. Agr. Res. 29,897–912. doi: 10.1071/AR9780897) rather than iTRAIT as the stress index. Indeed, the former expresses better the behaviour of each accession in response to the stress with respect to the mean behaviour the entire panel considered. I welcome comments by the authors about this point, even without introducing it in the text.

Yes, the SSI is popular when evaluating the susceptibility of genotypes to a given stress. The SSI and the index we used in the study (iTRAIT) are almost identical regarding the ranking of the individuals. The correlation between the two indices is equal to or smaller than -0.99 for all the traits. The main difference is the scale. As mentioned by the Reviewer, SSI makes it easier to rank the responses of the genotypes relative to the population mean. In the present study, using the SSI would have given the same results in terms of predictive ability. The choice of the iTRAIT over SSI was more a matter of work habits. Indeed, we regularly use this index, which gives us the relative performance in stress conditions. A negative value tells us directly about the impact of stress (not relative to the population).

-It is unclear why the authors excluded the possibility of evaluating the Na+ and K+ mass fraction of shoot tissue in the plants grown under the control condition (page 16, lines 354-355). In the control nutrient solution adopted (page 8, line 173; reference #41), the Na+/K+ molar ratio is about 0.18; thus, the Na+ and K+ mass fraction of shoot tissue could be detected. If the ion levels were below (unlikely) the detection limit of the ICP-AES technique adopted, an ICP-MS approach might be applied. Could the authors exclude that also for the NA/K trait the stress index had been used different results would have been obtained? Could the authors comment on this aspect?

The absence of measurement of Na+ and K+ mass fraction in the control condition was not related to technical constraints but to budget constraints. Given the budget available at the time of the experiment, we chose to have a more robust evaluation Na+ and K+ mass fractions in the salt condition rather than a less accurate evaluation in the two conditions. We assumed that the mass fractions in the control condition would not help to discriminate the genotypes in their response to stress. We recognized that is a limitation of the study.

It is difficult to say to which extent the results would have been different with the use of the index for these two traits (NA and K). It is well established that Na+ and K+ absorption in stress conditions is associated with tolerance to salinity (as highlighted in the present study with the ranking of tolerant/susceptible genotypes in the experiments). We, therefore, used the mass fractions in salt conditions to directly evaluate stress response. We can see that the predictions are much more correlated between the reference panel and the breeding population (Fig 6) in the salt condition. This is briefly highlighted in the discussion (Importance of validation on selection candidates). 

-Finally, other very minor points:

· What do the error bars in Fig.3 and 4 indicate? Please, introduce this information in the figure legends.

We added this information for the Fig 3. Fig 4 was modified to represent boxplots as suggested by Reviewer 1.

· Why the Na+/K+ ratios were evaluated from starting on ions mass faction and not as molar ratios?

As we reported the content of Na+ and K+ in mass fraction, we preferred to directly compute the ratio with the mass rather than mol.

---

## [Editor Report · Decision Letter 1]

6 Sep 2023

Genomic selection for salinity tolerance in japonica rice

PONE-D-23-12298R1

Dear Dr. Bartholomé,

We’re pleased to inform you that your manuscript has been judged scientifically suitable for publication and will be formally accepted for publication once it meets all outstanding technical requirements.

Kind regards,

Muhammad Abdul Rehman Rashid, PhD

Academic Editor

PLOS ONE

---

## [Editor Report · Acceptance letter]

18 Sep 2023

PONE-D-23-12298R1 

Genomic selection for salinity tolerance in *japonica* rice 

Dear Dr. Bartholomé:

I'm pleased to inform you that your manuscript has been deemed suitable for publication in PLOS ONE. Congratulations! Your manuscript is now with our production department. 

Kind regards, 

on behalf of

Dr. Muhammad Abdul Rehman Rashid 

Academic Editor

PLOS ONE